# Application of MODFLOW with Boundary Conditions Analyses Based on Limited Available Observations: A Case Study of Birjand Plain in East Iran

**Reza Aghlmand [1] and Ali Abbasi [1,2,]***

[1]  Department of Civil Engineering, Faculty of Engineering, Ferdowsi University of Mashhad, Mashhad 9177948974, Iran; rezaaghlmandcivil@gmail.com

[2]  Faculty of Civil Engineering and Geosciences, Water Resources Section, Delft University of Technology, Stevinweg 1, 2628 CN Delft, The Netherlands

*  Correspondence: a.abbasi@tudelft.nl or aabbasi@um.ac.ir; Tel.: +31-15-2781029

**Abstract:** Increasing water demands, especially in arid and semi-arid regions, continuously exacerbate groundwater resources as the only reliable water resources in these regions. Groundwater numerical modeling can be considered as an effective tool for sustainable management of limited available groundwater. This study aims to model the Birjand aquifer using GMS: MODFLOW groundwater flow modeling software to monitor the groundwater status in the Birjand region. Due to the lack of the reliable required data to run the model, the obtained data from the Regional Water Company of South Khorasan (RWCSK) are controlled using some published reports. To get practical results, the aquifer boundary conditions are improved in the established conceptual method by applying real/field conditions. To calibrate the model parameters, including the hydraulic conductivity, a semi-transient approach is applied by using the observed data of seven years. For model performance evaluation, mean error (ME), mean absolute error (MAE), and root mean square error (RMSE) are calculated. The results of the model are in good agreement with the observed data and therefore, the model can be used for studying the water level changes in the aquifer. In addition, the results can assist water authorities for more accurate and sustainable planning and management of groundwater resources in the Birjand region.

**Keywords:** groundwater modeling; GMS: MODFLOW; Birjand aquifer; calibration process

## 1. Introduction

Groundwater is a major source for drinking water, agricultural and industrial uses in arid and semi-arid regions. About 94.8% of Iran has an arid and semi-arid climate with low precipitation and high evapotranspiration rate and therefore, faces water scarcity [1,2]. It is estimated that around 98.7% of freshwater is available as groundwater [3].

Due to less vulnerability to pollution and high reliability, groundwater resources are commonly preferred for drinking water supply [4]. Groundwater is often not affected by short-term drought and therefore, can be considered as a reliable drinking water resource. However, it is difficult to obtain precise knowledge of aquifers because they are not visible like surface waters [5].

Groundwater models are the backbones of water resource planning and management in (semi) arid areas [6]. Nowadays, numerical modeling is considered an important tool for studying groundwater resources [7]. Generally, in groundwater models, a simplified mathematical representation of a groundwater system is solved by a computer program [8]. These models need varieties of

information—including geology, hydrogeology, hydrology, climatology, geography, etc.—to simulate the quantity/quality of the groundwater resources [9]. However, collecting such information, especially in developing countries, is a challenge and suffers from a high degree of uncertainty [10]. Quality of the input data in groundwater models has a significant effect on the model results. In other words, to get the accurate results, accurate input data should be ingested in the model [11]. Accordingly, the input data should be quality controlled and have the required resolution.

In this study, a three-dimensional, block-centered (cell-centered), steady-state, finite difference model, MODFLOW (McDonald and Harbaugh [12]) is employed to quantify groundwater in Birjand plain, South Khorasan, Iran. In recent years, GMS: MODFLOW model (Groundwater Modeling System) has been successfully developed and published in a large number of groundwater quantitative and qualitative studies because of its simple methods, modular program structure, and separate packages to resolve special hydrogeological problems [10,13–30]. This model, with a graphical user interface (GUI), can be integrated with geographic information system (GIS) to provide an appropriate visual environment for groundwater resources evaluation and management [15]. MODFLOW is considered an international standard for simulating and predicting groundwater conditions and groundwater/surface-water interactions [31]. Although MODFLOW has been applied for Birjand Plain in some literature, the real conditions—including source/sinks, recharges, extractions, return flows, soil coverage, etc.—have not been considered in detail. To fill these available gaps in the literature and previous studies conducted about Birjand aquifer, the boundary conditions as well as the input parameters in the model have been improved to reduce the bias of the simulated parameters such as hydraulic head distribution in the aquifer. To reach this aim, the limited available data are investigated and applied in the model effectively. Due to the lack of required data time series (such as head and flow), a semi-transient approach is applied to calibrate the parameters. In the GMS: MODFLOW, there are only two main approaches including steady-state and transient. Using a semi-transient approach allows consideration of the changes of the parameters in the study time period.

In the current research, the available data and the measurements are analyzed regarding their quality. Then, these data are prepared to use in the numerical model of Birjand aquifer. The data has been received from the Regional Water Company of South Khorasan (RWCSK), to construct the aquifer mathematical model.

In addition, the boundary conditions of the model are revised according to the available information. Using the measured values, the required parameters in the model are calibrated using a semi-transient method. Results show that the prepared model can be used in Birjand aquifer investigations and for predictions of the aquifer conditions under different development scenarios in the region.

## 2. Materials and Methods

### 2.1. Study Area

According to Iran Water Resources Management Company (IWRMC) [32], the number of deep and semi-deep wells utilized for extracting groundwater has been increased as shown in Figure 1. It should be noted that the shown numbers of wells in Figure 1 include only the authorized wells that have been licensed for exploitation. Unfortunately, the total number of groundwater extracting wells (either with license or without one) in the country is much higher than the available numbers in Figure 1. For the study region, the situation is the same. In Figure 2, the consumption of groundwater in different uses in South Khorasan province is shown. Groundwater is the main source of water supply for all types of uses in this region.

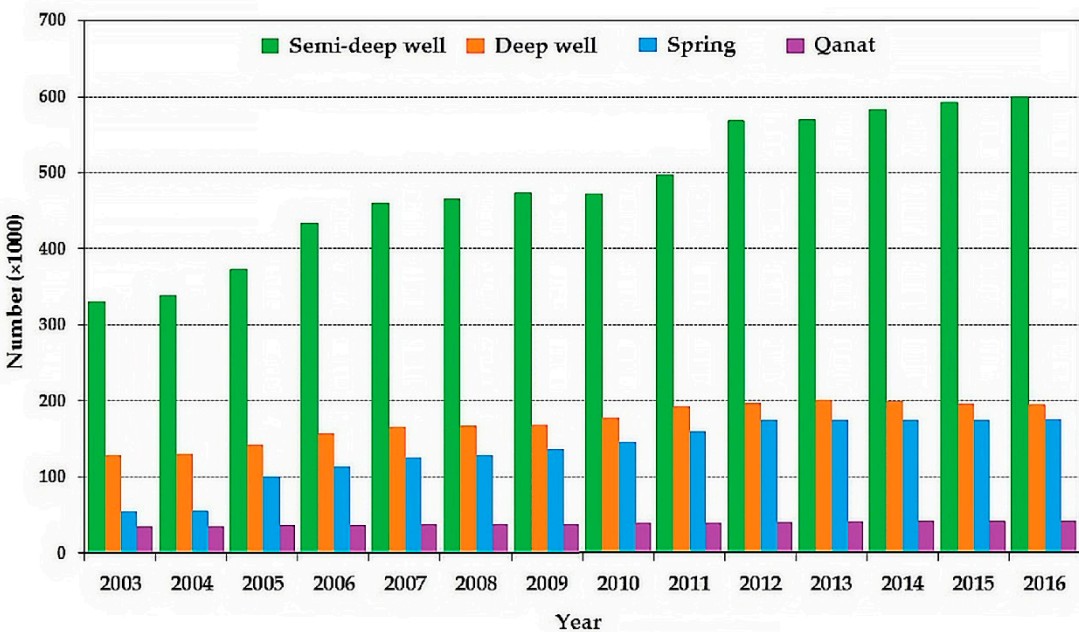

**Figure 1.** The number of deep and semi-deep wells, qanats, and springs in the period of 2003 to 2016 in Iran [32].

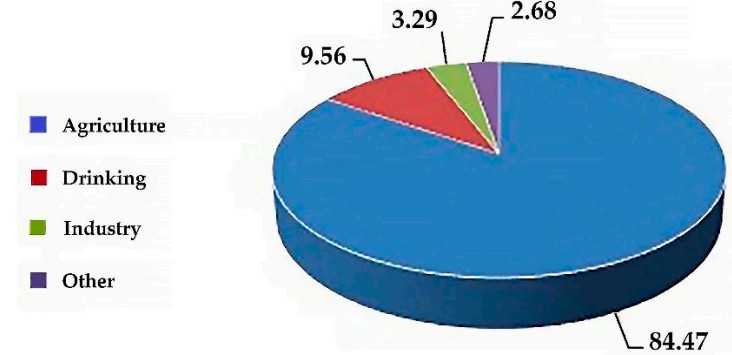

**Figure 2.** Percentage of groundwater use in different sections in South Khorasan province [33].

Birjand watershed, which includes the Birjand plain and the Birjand aquifer, is located in latitude and longitude of 32°36′ N to 33°8′ N and 58°41′ E to 59°44′ E, respectively. The location of the study region is shown in Figure 3. Birjand aquifer is located in an arid climate. The minimum, maximum, and the average temperatures recorded for the period 1989–2017 are −7.6, 38.3, and 16.6 °C, respectively. Due to the aridity of the region, the average annual precipitation is reported to be about 158 mm and hence, there is no perennial stream in this area. The slope of the ground surface gradually decreases from the eastern part toward the west. The western parts of the study area are almost flat as shown in Figure 3. The length of Birjand aquifer is about 55.0 km and the width in the middle is about 6.0 km. The average long-term temperatures in the eastern and western parts of the Birjand watershed are about 14 and 16 °C, respectively. Also, the average long-term annual precipitation in the easternmost part is about 160 mm, while in the westernmost part of the desired area is about 120 mm. Minimum and maximum annual evaporation occur in the easternmost part (2200 mm) and westernmost part of the watershed (3400 mm), respectively. Due to over-exploitation of Birjand aquifer through both authorized and unauthorized wells in the study area, this aquifer has been declared a prohibited aquifer. About 80% of groundwater discharge in the study area occurs through deep and semi-deep wells and the rest occurs through springs and qanats [34].

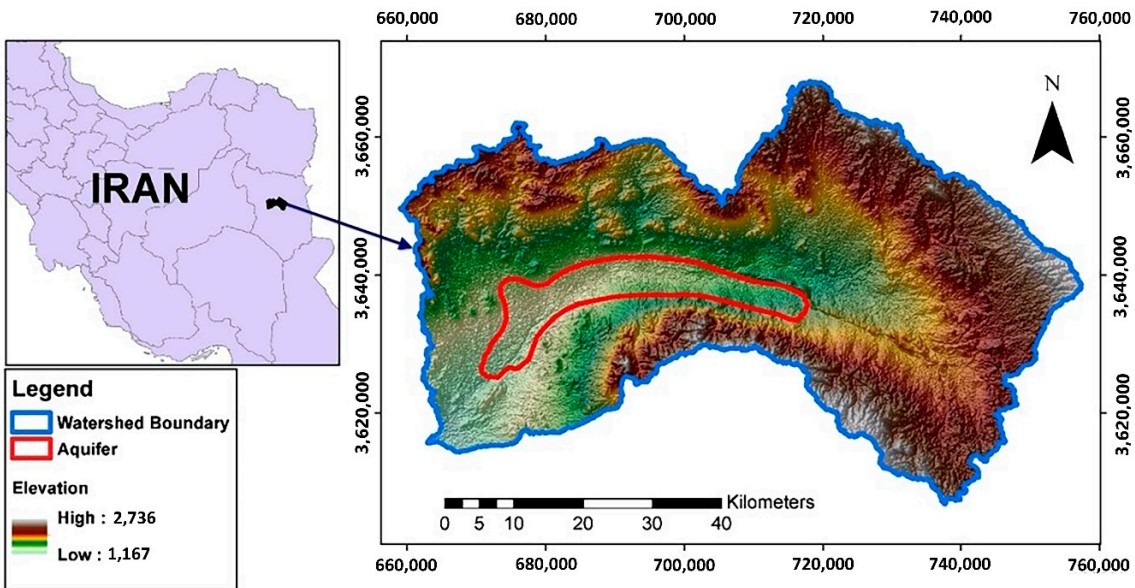

**Figure 3.** Location of Birjand watershed and Birjand Aquifer in South Khorasan Province, Iran [35].

The groundwater level in Birjand plain, like most other plains in Iran, are continuing to decline due to long-term droughts and excessive extraction, especially by the agriculture sector. Over a 30-year period from 1987 to 2018, the average monthly groundwater level has dropped from 1354.22 m to 1340.51 m (i.e., drawdown about 14.0 m). It means that the water level in Birjand aquifer, as the main source of water supply in the city of Birjand, has declined annually by an average of about 0.45 m over the past 30 years (Figure 4). Consequently, the total deficit of the Birjand groundwater reservoir has been 193.63 million cubic meters (MCM) with an average annual deficit of 6.45 MCM. This situation shows that the Birjand plain has been in a severe water crisis [36].

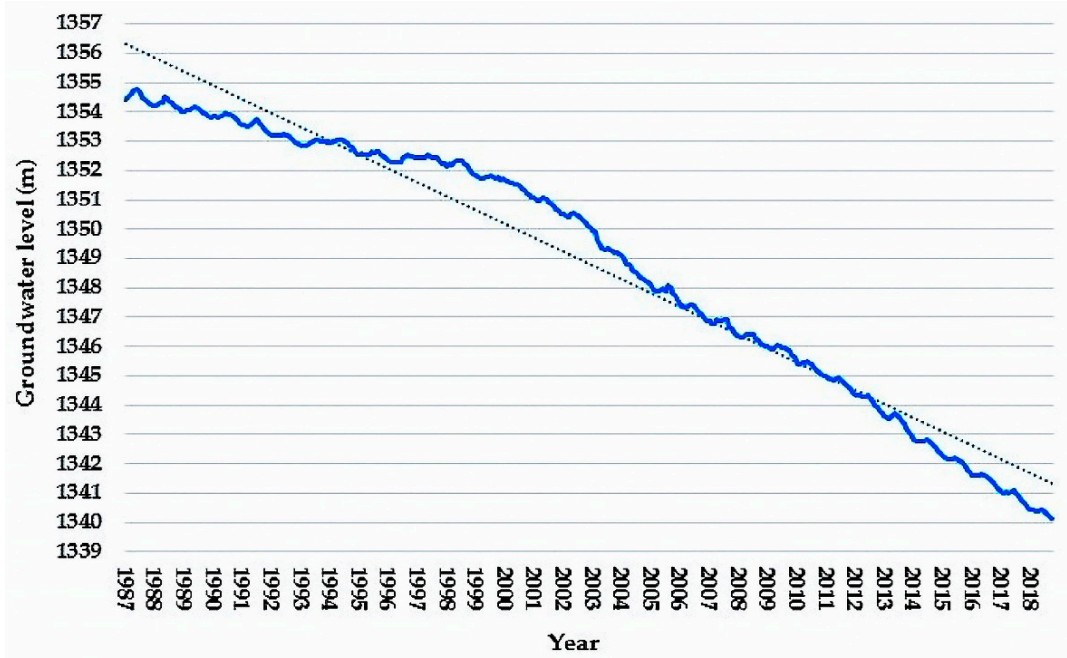

**Figure 4.** Groundwater hydrograph of Birjand Plain during a 30-year period.

Birjand aquifer is unconfined and in the context of climate change, unconfined aquifers in arid and semi-arid areas compared with ones in wet/rainy areas are more vulnerable. The reason is that the occurrence of droughts in arid and semi-arid areas exacerbates the aquifer's condition through

decreasing in aquifer recharge [37]. This factor, along with other factors such as population growth and increasing demand, leads to a continuous decrease in groundwater resources in these areas.

Investigating the geological maps of the Birjand plain area shows that the Birjand aquifer structure totally is related to the quaternary formation, dating from the quaternary period. The quaternary period began about two and a half million years ago and continues today, and is, in fact, the newest geological period. Therefore, from a geological point of view, the whole body of the Birjand aquifer consists of young deposits (Figure 5). The young quaternary deposits include sediments that are eroded and deposited by rivers in this area and are generally coarse-grained rock material. It should be noted that the young quaternary deposits that cover the whole Birjand aquifer have the highest share (about 25%) among the lithological units of the entire Birjand watershed.

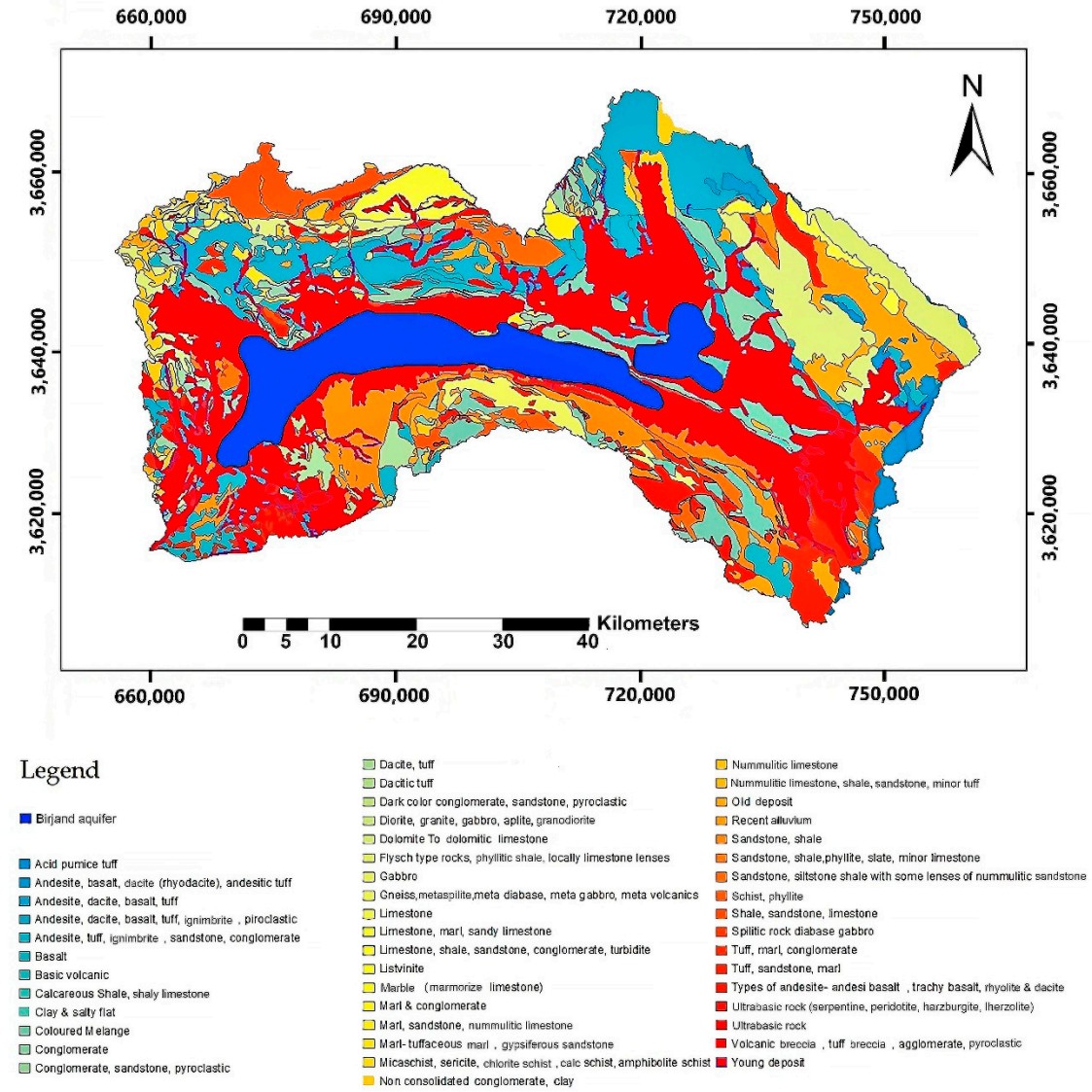

**Figure 5.** Birjand watershed geological map.

Regarding the soil properties, the southern, eastern, and northern parts of the Birjand watershed are composed of high and relatively high mountains consisting of limestone, metamorphic rock, conglomerate, sandstone, and shale rocks. These areas are generally vegetation-free or with little vegetation. The central parts of the Birjand watershed are related to lowlands with relatively mild slopes. The central parts mainly have very deep soils, consisting of sedimentary rivers. The western part of the Birjand watershed also has relatively high hills, consisting of metamorphic rocks and

shale in the north-facing part; and in the central- and south-facing parts: limestone, dolomitic, and sandstone—generally coarse-grained. These areas are generally vegetation-free or very low on vegetation and have low/very low-deep soils.

As seen in Figure 6, there are two aquifers in the Birjand watershed including Birjand aquifer and Marak aquifer. The areas of the Birjand and Marak aquifers are 277.8 km$^2$ and 53.72 km$^2$, respectively. These two aquifers are separated from each other due to the rising bedrock elevation between them.

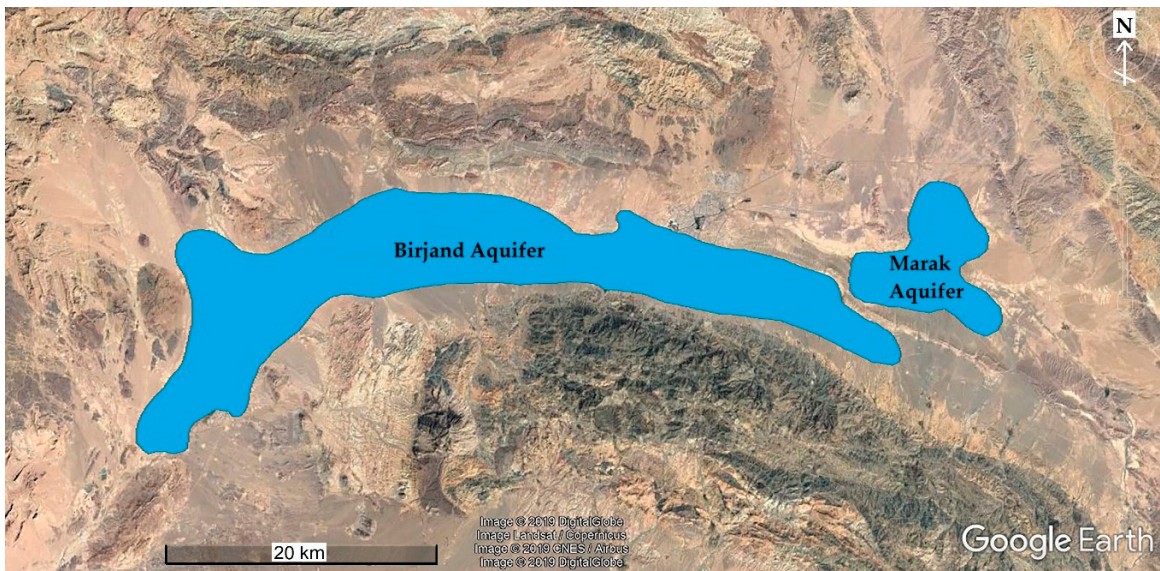

**Figure 6.** The study area and the exact location of Birjand and Marak aquifers.

## 2.2. Groundwater Modeling of Birjand Aquifer

Understanding a groundwater system usually requires drilling a large number of exploratory wells, drilling, and pumping operations, and conducting multiple geophysical experiments and a series of long-term experiments, which are expensive and time-consuming. Unfortunately, in the study area, very few field operations and surveys have been carried out and therefore, modeling the groundwater flow through a mathematical model can be very promising.

There are some groundwater modeling programs which were developed on the basis of various methods. The most famous models (GUIs) are Visual Modular Three-Dimensional Finite-Difference Flow Model (Visual MODFLOW) [38], Finite Element Subsurface Flow System (FEFLOW) [39], Groundwater Modeling System (GMS) [40], etc. FEFLOW, which uses the finite element method for modeling, and GMS and Visual MODFLOW are the most popular software packages applied in groundwater studies [14,19,41–52].

The GMS software is a graphical user interface for many groundwater models such as FEMWATER, SEEP2D, SEAM3D, MT3DMS, MODFLOW (with many packages), RT3D, MODPATH, MODAEM, and SEAWAT. In this study, the MODFLOW model has been chosen due to its high efficiency and its extensive use in groundwater studies. This model simulates the flow in three dimensions using finite difference method for both steady-state and transient conditions. The MODFLOW numerical model is constructed based on the combination of two basic equations—the Darcy equation and the principle of conservation of mass, or mass balance.

Governing Equations

The three-dimensional groundwater flow with constant density through a heterogeneous and anisotropic porous medium can be described by the equation [12]

$$\frac{\partial}{\partial x}\left(K_{xx}\frac{\partial h}{\partial x}\right) + \frac{\partial}{\partial y}\left(K_{yy}\frac{\partial h}{\partial y}\right) + \frac{\partial}{\partial z}\left(K_{zz}\frac{\partial h}{\partial z}\right) \pm W = S_s\frac{\partial h}{\partial t} \tag{1}$$

where $K_{xx}$, $K_{yy}$, and $K_{zz}$ are hydraulic conductivity coefficients (L/T) in $x$, $y$, and $z$ directions, respectively; $h$ is the pressure head (L); $S_s$ is specific storage (1/L); and $W$ is recharge/discharge rate per unit volume (1/T). The environment is unconfined, isotropic, and heterogeneous ($K_{xx} = K_{yy} = K_{zz} = K$), and hence, the governing equation based on Dupuit assumptions [53] in two-dimensional form can be written as

$$\frac{\partial}{\partial x}\left(Kh\frac{\partial h}{\partial x}\right) + \frac{\partial}{\partial y}\left(Kh\frac{\partial h}{\partial y}\right) \pm W = S_y\frac{\partial h}{\partial t} \tag{2}$$

where $S_y$ is the specific yield (dimensionless).

Due to the lack of long-term monitoring data for observational/operational wells, amounts of inflow/outflow to/from the Birjand aquifer are unknown and therefore, the simulations are limited to steady-state conditions. Although applying the steady-state groundwater model simulation in Birjand aquifer is a forced choice, we are trying to consider the real-world conditions in the modeling.

## 2.3. Groundwater Conceptual Model of Birjand Aquifer

The first and most important step in groundwater modeling is constructing the conceptual model of the groundwater system [54], which represent a simplified version of the actual aquifer system. Due to the complexity of the hydrogeological system, as well as the lack of data in the study region, the conceptual model and its structure is applied according to the available data [55]. Establishing the groundwater conceptual model in the study region suffers from many challenges including:

- Lack of adequate knowledge and incomplete information about the physical properties of alluvial deposits of plain, which is the main reservoir of groundwater;
- Lack of adequate and accurate statistics and information on meteorological and climatic parameters and other parameters in the study area for estimating the water balance components;
- Lack of sufficient observation wells and other observations in the area;
- Lack of accurate and adequate statistics and criteria on the method and extent of utilization of Birjand groundwater resource;
- Lack of sufficient exploratory wells in the study plain to understand the physical and geometric characteristics of the aquifer;
- Error in piezometer recorded values;
- Lack of adequate pumping tests in the study area and therefore, lack of sufficient information on the hydrodynamic coefficients of the aquifer;
- Lack of sufficient understanding of the hydraulic behavior of an aquifer's surrounding formations and their relationship with the aquifer, and the consequent lack of proper and precise definition of boundary conditions;
- Lack of sufficient information on the hydraulic connections between surface water (e.g., river or lake) and groundwater resources;
- Lack of sufficient information for calculating agricultural, urban, and industrial backwaters;

The different steps for developing the conceptual model of the Birjand aquifer are described in Figure 7.

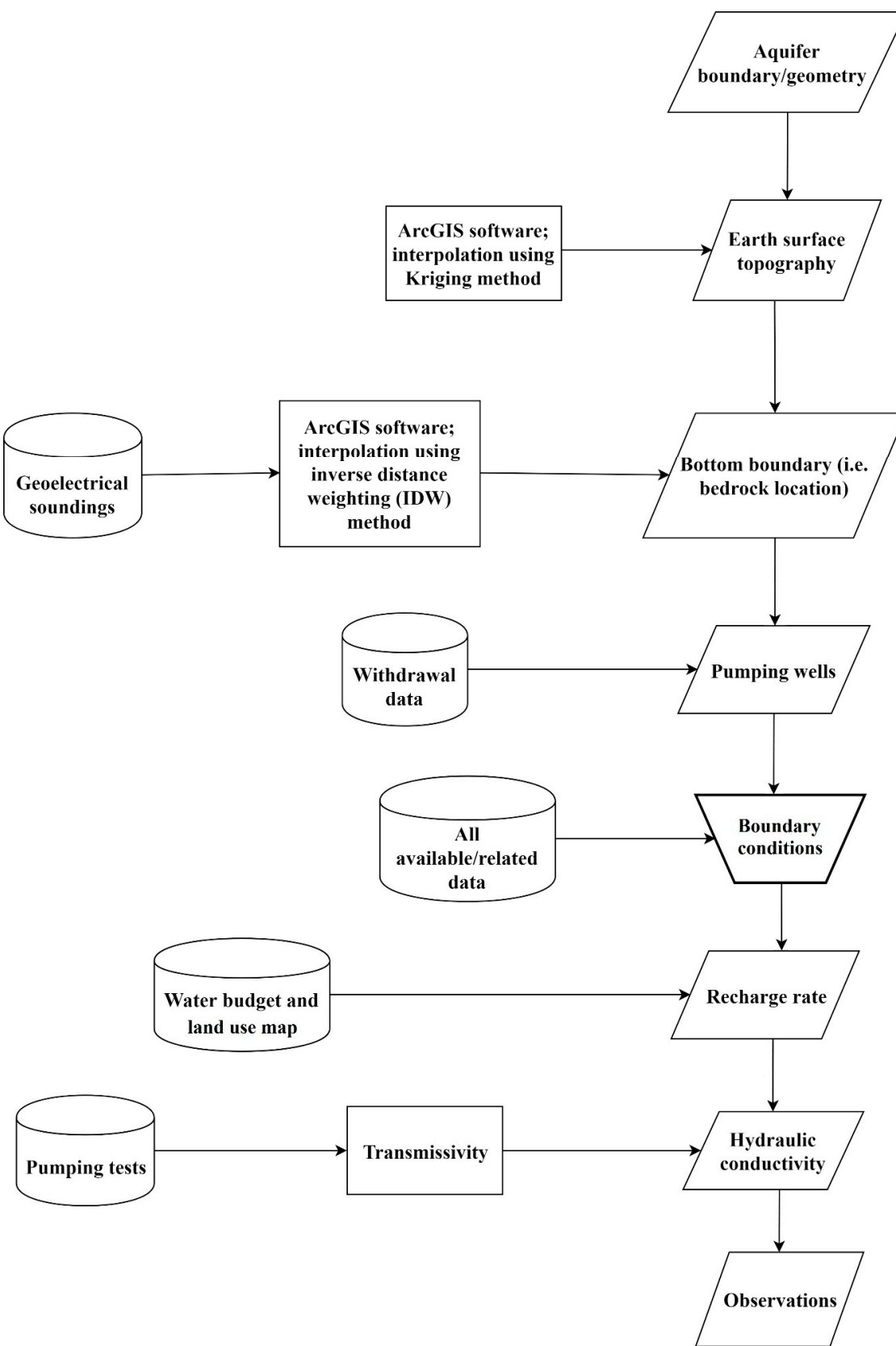

**Figure 7.** Flowchart of the building the conceptual model of Birjand aquifer.

The aquifer geometry is determined according to the RWCSK data. A comprehensive study (e.g., lithology and geology studies) has been done in this research to be able to model the aquifer

boundaries (i.e., aquifer geometry) accurately in the model [35,56–62]. The bottom boundary (i.e., the bedrock in the aquifer) is also determined using the limited available geophysical study (geoelectrical soundings) carried out in Birjand Plain in 1971 (Figure 8).

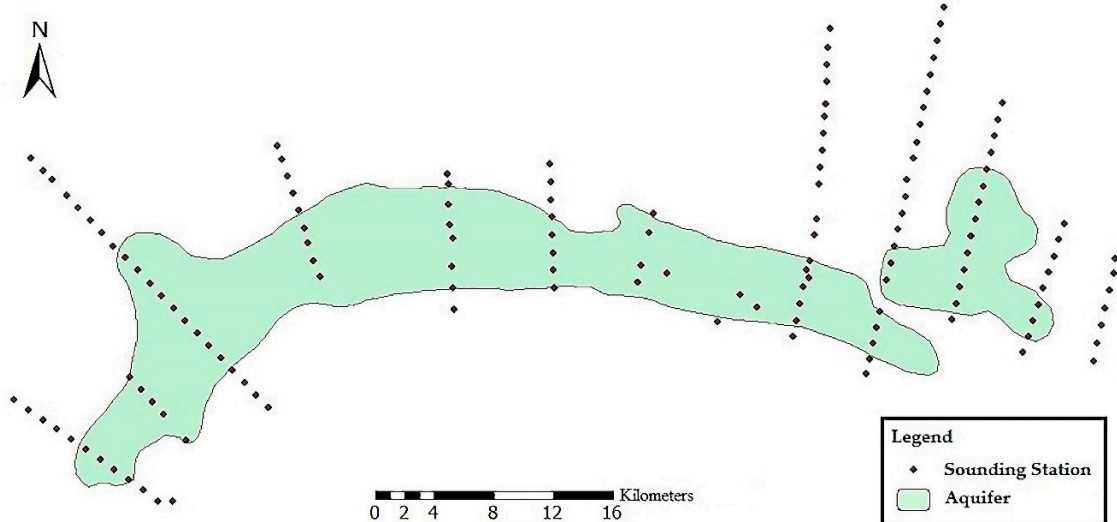

**Figure 8.** Location of geoelectrical soundings in Birjand aquifer.

To define the sinks and sources in the aquifer model, the positions and amounts of withdrawals/discharges from each well should be determined. There are 187 pumping/extracting wells with recorded data in the study area which are applied in the model (Figure 9). The well data are obtained from RWCSK and their quality are controlled before importing to the model. Of the total wells used in the model, 26 wells are used for drinking water and sanitation, 9 wells for livestock, 26 wells for industry and services, 119 wells for agriculture, and 7 for other uses.

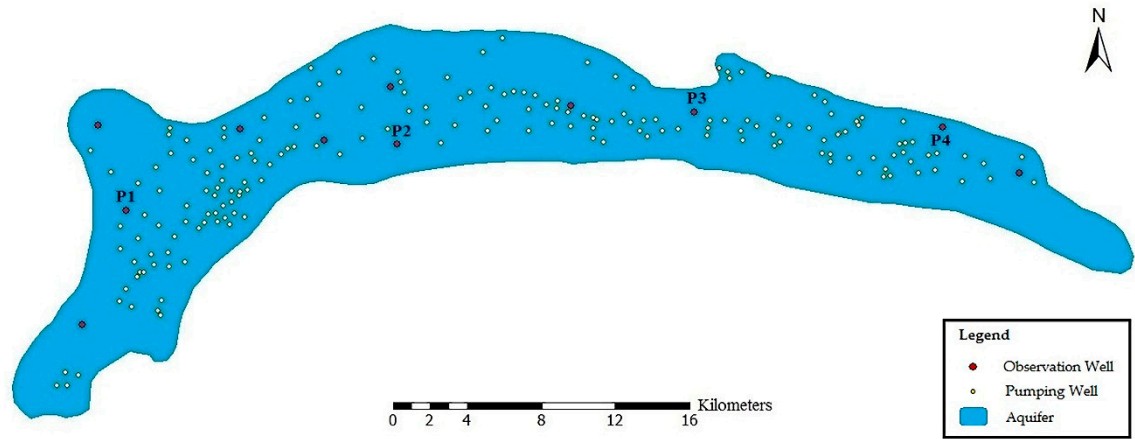

**Figure 9.** Location of pumping and observation wells (piezometers) in Birjand aquifer. P1, P2, P3, and P4 are for selected piezometers to check the groundwater level changes at the end of calibration and verification processes.

### 2.3.1. Boundary Conditions

The governing equations of the groundwater flow are solved using the finite difference approach. This requires that the boundary and initial conditions of the problem are described in details in the model [63]. The groundwater budget components of the Birjand aquifer provided by RWCSK are presented in Table 1. This budget can present a good view of the overall status of sinks and sources in the aquifer.

**Table 1.** Birjand aquifer budget provided by RWCSK [64]. The values are in million cubic meters (MCM) per year.

| Inputs (MCM/Year) | | Outputs (MCM/Year) | |
|---|---|---|---|
| Lateral underground inflow | 25.44 | Discharge and extraction (well, qanat, spring) | 73.56 |
| Infiltration of precipitation | 4.08 | Lateral underground outflow | 1.15 |
| Infiltration of runoff | 4.47 | Drainage | 0.00 |
| Infiltration of agricultural wastewater | 17.87 | evapotranspiration | 0.00 |
| Infiltration of drinking and industrial wastewaters | 14.32 | The total volume of discharge | 74.71 |
| The total volume of recharge | 66.18 | | |

In previous studies, different types of boundary conditions have been used in Birjand aquifer. In the present study, we are trying to apply precise aquifer input/output boundaries using geological surveys and satellite imagery through the Google Earth Pro software.

To find the real-world conditions, a comprehensive investigation of the adjacent areas of the aquifer, geological conditions, lithology maps, the type of soils, and topography of the area has been done. Finally, by using this data, the lateral input boundaries or lateral recharge are defined in the model as follows:

- In the northeast part of the aquifer, there is an exchange of groundwater between Birjand and Marak aquifers (Figure 6). The groundwater flow direction in this area is from the Marak aquifer towards Birjand aquifer (the water levels are higher in the outlet of the Marak aquifer) with the flow rate of about 3.56 million cubic meters per year.
- The second input boundary area to the Birjand aquifer located in the south as shown in Figure 10. The southern parts of the Birjand aquifer have the highest elevations of the land structure in the adjacent areas of the Birjand aquifer. In addition, there are alluvial fans as unconsolidated sedimentary deposits in these parts which, due to having steep slopes, can recharge the aquifer during the precipitation.

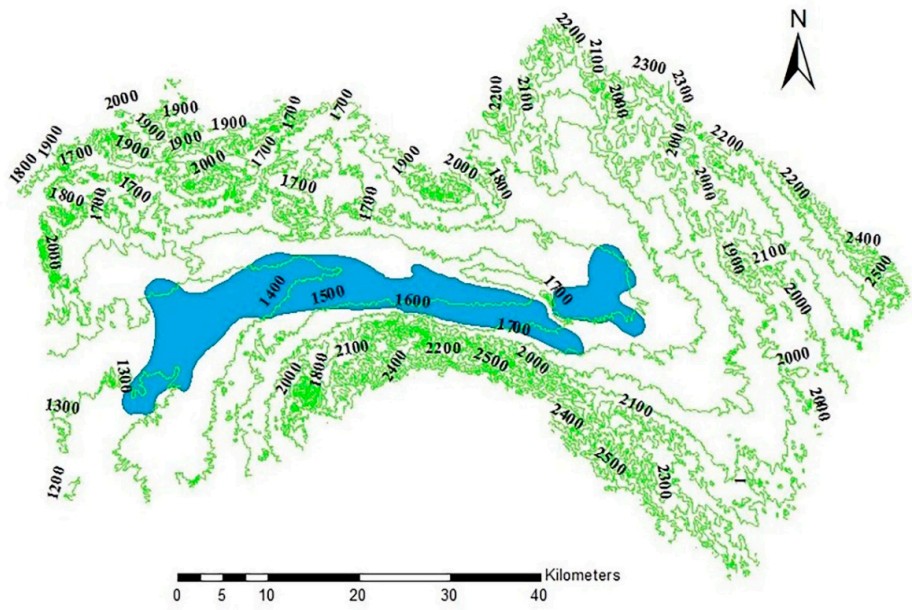

**Figure 10.** Topography of the Birjand watershed which includes the Birjand aquifer. The numbers represent the elevation in m.s.l.

- There is another lateral input boundary in the northwest of the aquifer, which looks like a camel hump. In this area, there is a large fan-shaped alluvial cone that has been washed out or eroded over the years from high altitudes and dispersed in a large area with a perimeter of about 17.5 km, as shown in Figure 11. Precipitation over this alluvial cone flows through specified paths and then penetrates into this vast area and joins the Birjand aquifer.

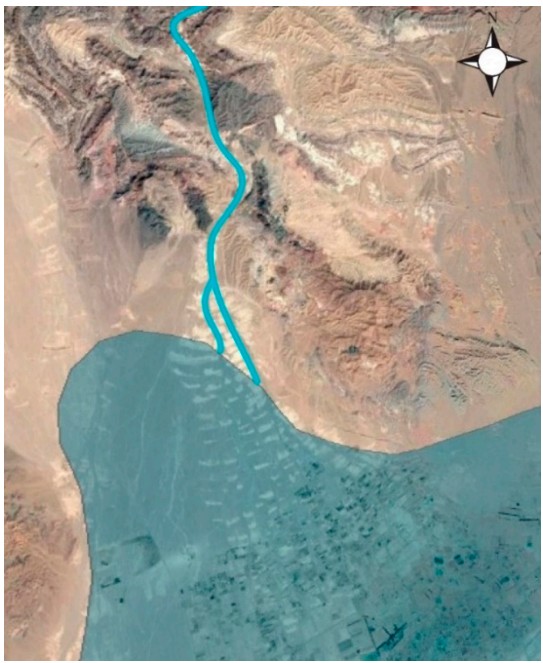

**Figure 11.** Large alluvial fan as an input boundary or lateral recharge, located in the northwest of Birjand aquifer. The blue line shows the groundwater lateral recharge way.

- Lithology, as well as soil type and soil texture investigations in the study region are identified the fourth input boundary. In Figure 6, the southern part of the aquifer and from the central side towards the west of the aquifer, geologically are formed from predominantly sandstone, siltstone, phyllite, slate, and minor limestone, which have very low permeability. As a result, due to the fact that water cannot penetrate rapidly, it becomes runoff and flows downstream, and penetrates as soon as it enters the aquifer's alluvial zone, causing the aquifer recharge. The relatively high slope in this area, which speeds up the runoff from precipitation, as well as the presence of a large mountain above this rock that has average precipitation above the Birjand plain, are some factors that help to recharge the aquifer. The distance between the second and fourth major input areas (both of which are located in the southern part of the Birjand aquifer) mainly is not considered as an input boundary because there are relatively low elevations and slight/gentle slopes between these elevated areas in the southern part. In other words, in the area between these two inputs, the stone structure is far from the aquifer boundary. Due to aridity of this region (high evapotranspiration and small precipitation), the amount of water that penetrates in this area is not transported to the Birjand aquifer.
- There is another input boundary in the northern areas of the Birjand aquifer. There are two large alluvial fans in this area, with a tip distance of about 7.2 km (Figure 12) and at the base, these are located entirely within the aquifer boundary and their distance is reduced by about half. These alluvial fans build a place for penetrating the runoffs to the aquifer and recharging it. In the upstream part of the right (eastern) alluvial fan, there are some human activities, such as leveling the ground, constructing a small earth dam, and farming. Therefore, input flows from this side can be ignored and considered as a no-flow boundary. However, the upstream of the left alluvial fan (western) remains almost virtually undisturbed without any considerable human activities.

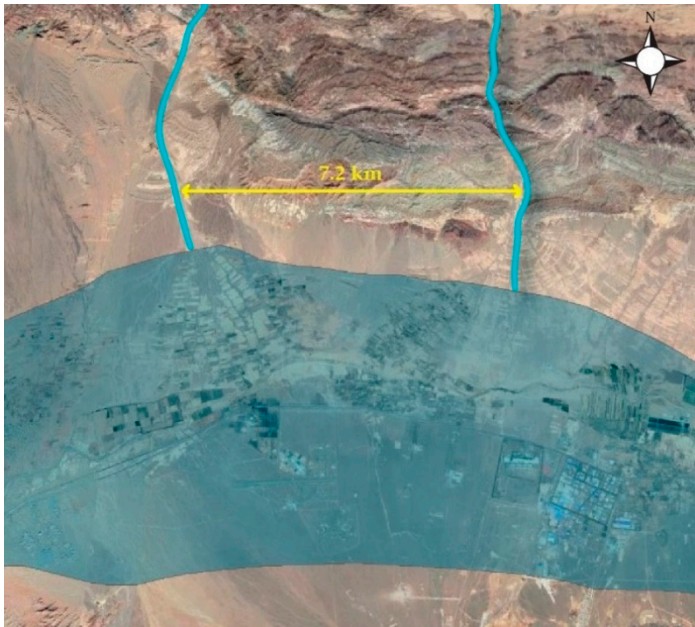

**Figure 12.** Alluvial fans position at the northern boundary of the Birjand aquifer. The blue line shows the groundwater lateral recharge way.

The above mentioned five main input areas are considered as specified head boundary or Dirichlet/first-type boundary conditions in the model. The rest boundaries are considered as no-flow boundaries or Neumann/second-type boundary conditions because there are no hydraulic connections between the aquifer and its neighbors (Figure 13).

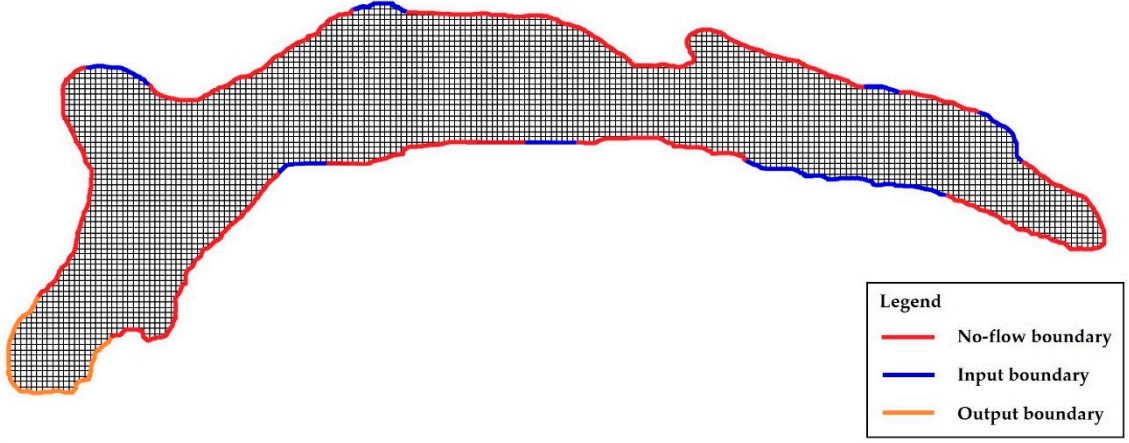

**Figure 13.** Boundary conditions of the model.

It is assumed that the differences between the ground's surface gradient and the groundwater level gradient are small and therefore, each input boundary's head can be calculated based on the head of the adjacent observation well. Regarding this issue, the distance between the input boundary and adjacent observation well should not be too high.

### 2.3.2. Model Parameters

One of the most important parameters in a groundwater model is hydraulic conductivity. The hydraulic conductivity values can be calculated based on the transmissivity values (*T*) obtained from the pumping tests using the equation

$$K = T/B \tag{3}$$

where *K* represents hydraulic conductivity (m/d); *T* is transmissivity (m$^2$/d); and *B* shows the thickness of the aquifer saturation layer (the difference between groundwater level and bedrock level in each point) (m). Finally, using inverse distance weighting (IDW) interpolation method in GMS, an approximate initial value of the aquifer hydraulic conductivity is obtained.

Calculating the aquifer recharge often is one of the most challenging issues in groundwater studies. According to Table 1, the total amount of aquifer recharge—including the penetration of precipitation, surface runoff, agricultural, drinking, and industry wastewaters—is about 40.74 MCM per year. In this study, this amount of direct recharge distributed in the aquifer conceptual model based on the land use map.

### 2.3.3. Model Computational Grid

The results of the groundwater model depend on the size of computational grids. In this study, as shown in Figure 14, the grid cell size is 250 × 250 m in horizontal plane uniformly and the height of each cell is equal to alluvial depth in that point (i.e., the difference between earth's surface and bedrock levels). The modeling grid consists of 79 rows and 224 columns. The total number of grid cells is 17,696, including 4088 active cells (all cells inside the aquifer are active) and 13,608 inactive (all cells outside the aquifer are inactive).

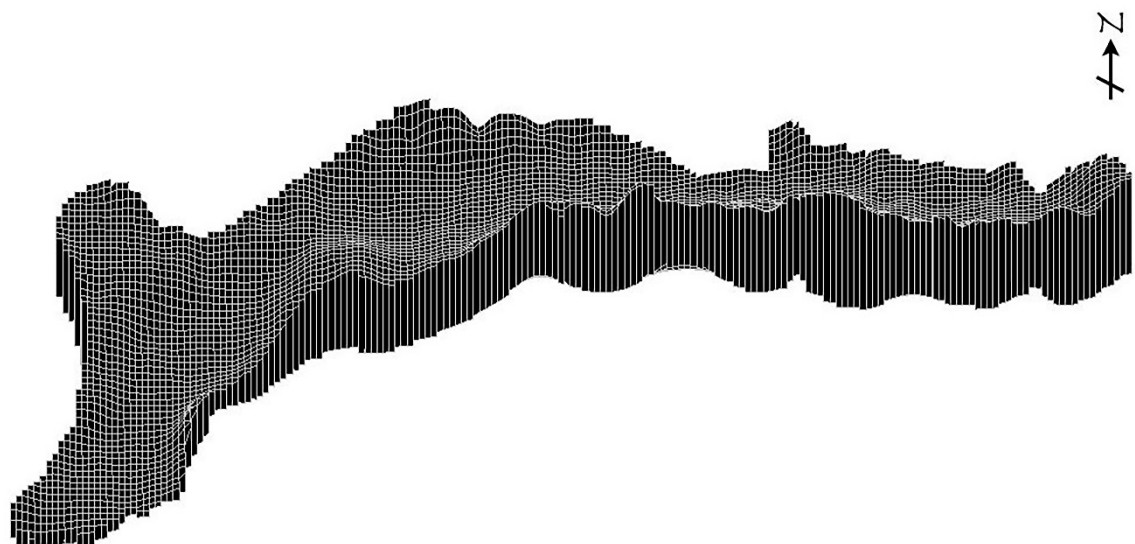

**Figure 14.** 3-D grid created by GMS: MODFLOW for Birjand groundwater model.

## 3. Results

### *3.1. Model Calibration*

There are generally two kinds of calibration process; the first one is a trial-and-error process that should be manually changed repeatedly calibration parameters. This method can be considered as a fundamental first step for history matching because it can give the modeler much insight about the site modeled and how parameter changes affect different areas of the model and types of observations [65]. The second type is automated parameter estimation which in many cases can calibrate the model quickly.

GMS contains an interface to the mentioned calibration called PEST (Parameter ESTimation) [66]. PEST calibration can be performed in two ways including zonal and pilot point. The first approach (i.e., zonal) is the most common one [67] and is applied in this study.

For calibration, the hydraulic head data of 11 observation wells or piezometers in the study region is imported to model. Using the trial-and-error approach, attempts are made to minimize the differences between calculated and observed head values. The quality of the calibration is evaluated using some indices including mean error (ME), mean absolute error (MAE), and root mean square error (RMSE) according to the equations

$$\text{ME} = \frac{1}{n} \sum_{i=1}^{n} (h_o - h_c)_i \tag{4}$$

$$\text{MAE} = \frac{1}{n} \sum_{i=1}^{n} \left| (h_o - h_c)_i \right| \tag{5}$$

$$\text{RMSE} = \left[ \frac{1}{n} \sum_{i=1}^{n} (h_o - h_c)_i^2 \right]^{0.5} \tag{6}$$

where $n$ is the number of piezometers; $h_o$ and $h_c$ show observed/measured and calculated/simulated head values (m), respectively. Calculation of the above mentioned statistic indices is useful in evaluating the merit of the calibration [68]. It should be noted that the GMS software provides ME, MAE, and RMSE values for each model run. Because both positive and negative residuals are used in calculation, ME value should be close to zero for a good calibration. MAE is calculated using the absolute values of the error (only positive values) and is a measure of the average error in the model. The root mean square error (RMSE) or the standard deviation (RMSD) (due to using the steady state results in calibration RMSE and RMSD are equal) is the average of the squared differences in measured and simulated heads. RMSE is less robust to the effects of outlier residuals. Thus, the RMSE is typically larger than the MAE.

Calibration of the model is performed for steady-state in this study because the time series of groundwater flow are unknown. Instead of transient calibration, a semi-transient calibration approach is applied for Birjand aquifer. In this way, calibration of the model is carried out for seasonal data of the study period (about 7 years) as summarized in Table 2. The main reason for choosing a season as a time step is that the groundwater level changes in all observation wells were insignificant during a season. The average groundwater levels in each piezometer during each season were considered and entered to the model. The results show that there is a good agreement between the input data, calibrated parameters, and the assumption in the study period, and the model can be used beyond this time period.

Due to the importance of hydraulic conductivity, model calibration can be used for determining this parameter. Regarding this issue, the Birjand aquifer is divided into 25 polygons or zones as shown in Figure 15.

To illustrate the difference between observed and calculated head values during the study period, four piezometers were chosen randomly according to Figure 16. The locations of these four selected piezometers are shown in Figure 9. As seen in Figure 16, the groundwater level almost continuously decreases in all selected piezometers in the study area.

**Table 2.** Semi-transient calibration of Birjand groundwater model.

| Season | Mean Error (m) | Mean Absolute Error (m) | Root Mean Square Error (m) |
|---|---|---|---|
| Spring 2018 | −0.04 | 0.14 | 0.18 |
| Winter 2018 | 0.03 | 0.14 | 0.19 |
| Autumn 2017 | 0.00 | 0.16 | 0.22 |
| Summer 2017 | 0.03 | 0.17 | 0.20 |
| Spring 2017 | 0.13 | 0.17 | 0.23 |
| Winter 2017 | 0.14 | 0.19 | 0.25 |
| Autumn 2016 | 0.05 | 0.22 | 0.28 |
| Summer 2016 | 0.02 | 0.26 | 0.30 |
| Spring 2016 | 0.09 | 0.22 | 0.27 |
| Winter 2016 | 0.09 | 0.23 | 0.27 |
| Autumn 2015 | 0.04 | 0.27 | 0.33 |
| Summer 2015 | 0.05 | 0.28 | 0.32 |
| Spring 2015 | 0.16 | 0.26 | 0.35 |
| Winter 2015 | 0.15 | 0.27 | 0.31 |
| Autumn 2014 | 0.05 | 0.27 | 0.30 |
| Summer 2014 | 0.03 | 0.27 | 0.30 |
| Spring 2014 | 0.14 | 0.26 | 0.28 |
| Winter 2014 | 0.12 | 0.25 | 0.27 |
| Autumn 2013 | 0.04 | 0.27 | 0.30 |
| Summer 2013 | 0.13 | 0.28 | 0.32 |
| Spring 2013 | 0.08 | 0.25 | 0.27 |
| Winter 2013 | 0.03 | 0.27 | 0.30 |
| Autumn 2012 | 0.01 | 0.24 | 0.31 |
| Summer 2012 | −0.01 | 0.25 | 0.31 |
| Spring 2012 | 0.14 | 0.23 | 0.30 |
| Winter 2012 | 0.14 | 0.24 | 0.31 |
| Autumn 2011 | 0.06 | 0.24 | 0.33 |
| Summer 2011 | 0.06 | 0.25 | 0.33 |
| Spring 2011 | 0.13 | 0.23 | 0.31 |

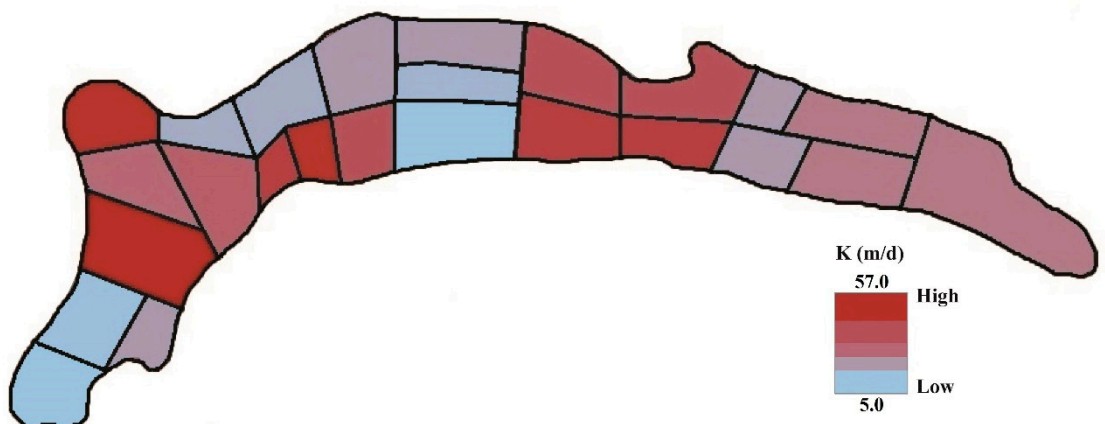

**Figure 15.** Distribution of hydraulic conductivity values in Birjand aquifer.

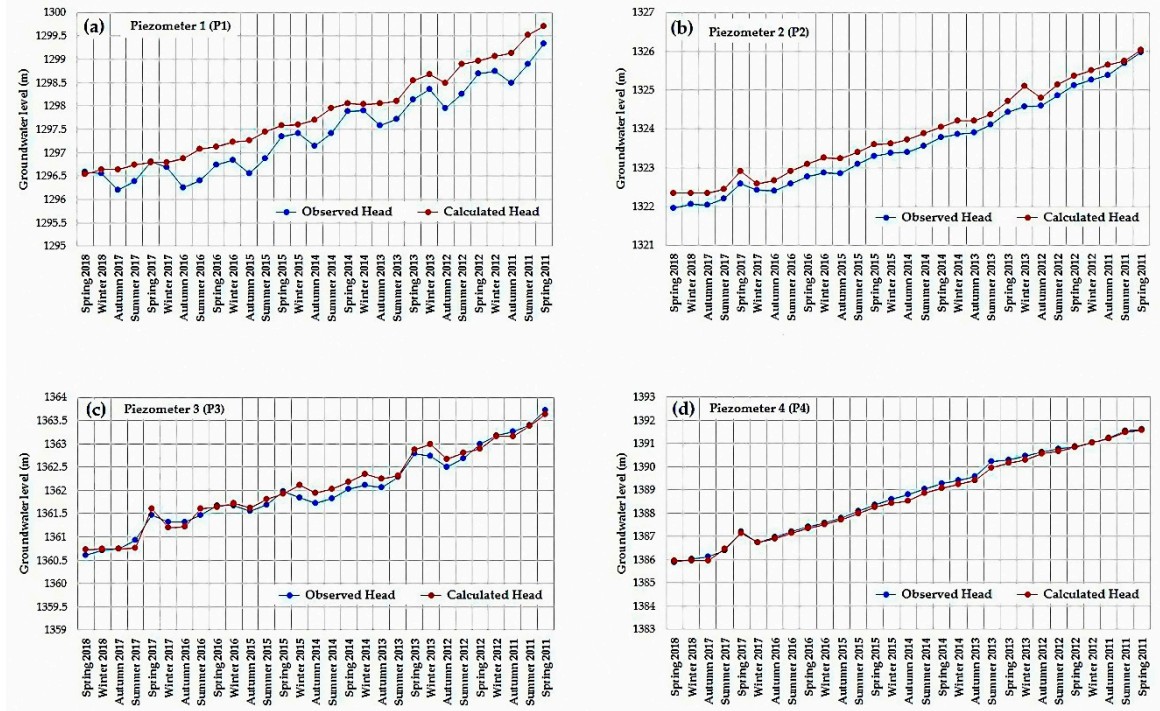

**Figure 16.** The results of semi-transient calibration of the model, Birjand groundwater level changes as well as observed and calculated head values for the four selected piezometers include (**a**) piezometer 1; (**b**) piezometer 2; (**c**) piezometer 3; and, (**d**) piezometer 4, over a period of about 7 years as seasonal.

As shown in Figure 9, all four piezometers are selected randomly from different areas of the aquifer to investigate groundwater levels. Accordingly, it is clear that P1 and P2 have a relatively big difference between observed and calculated head values. The reason is that these two piezometers are located in the western half of the aquifer. It should be noted that after investigating calibration results for all available piezometers and for all the seasons, we noticed that the difference between the observed and calculated heads in the piezometers of the western parts of the aquifer were a bit higher than the eastern piezometers. Investigation of observation wells data during different months and seasons showed that the temporal variation of the piezometers head values in the western half of the aquifer was more slowly than the piezometers in the eastern half. In other words, the groundwater level decrease over time in the western parts of the aquifer occurs more slowly than the eastern parts; Analysis of piezometer statistics showed that the average annual groundwater level drop in western piezometers was about half of the water level drop in eastern piezometers. Therefore, simultaneous calibration of the entire aquifer is a challenging issue and in this case, it is almost impossible to make the difference between the observed and calculated heads across all piezometers near to zero.

In the present study, all error values reported for model calibration (and model evaluation in the next section) indicate mean values for each error at each season and these errors are general errors of the calibrated/evaluated model that derived from all available piezometers.

### 3.2. Model Evaluation

After the calibration process, the prepared model should be evaluated to prove the model is reliable in different conditions. In this section, the calibrated parameters for the most recent time (spring 2018) is chosen for evaluating the model results. As shown in Table 3, the agreement between the results of the model and the measurements is promising for both parameters including hydraulic conductivity (K) and recharge (R) and consequently, for the model. Similar to the calibration section, the groundwater level changes in the same four selected piezometers for model evaluation over 7 years are shown in Figure 17. Each red point represents the amount of the head difference and belongs to a

spring season of a specific year that is specified beside each point. P1, P2, P3, and P4 are the selected piezometers as shown in Figure 9.

**Table 3.** Evaluation results of Birjand groundwater model over a 7-year period.

| Season | Mean Error (m) | Mean Absolute Error (m) | Root Mean Square Error (m) |
|---|---|---|---|
| Spring 2017 | 0.06 | 0.13 | 0.20 |
| Spring 2016 | 0.05 | 0.19 | 0.24 |
| Spring 2015 | 0.06 | 0.20 | 0.27 |
| Spring 2014 | 0.05 | 0.18 | 0.23 |
| Spring 2013 | 0.03 | 0.19 | 0.24 |
| Spring 2012 | −0.05 | 0.22 | 0.28 |
| Spring 2011 | −0.03 | 0.22 | 0.30 |

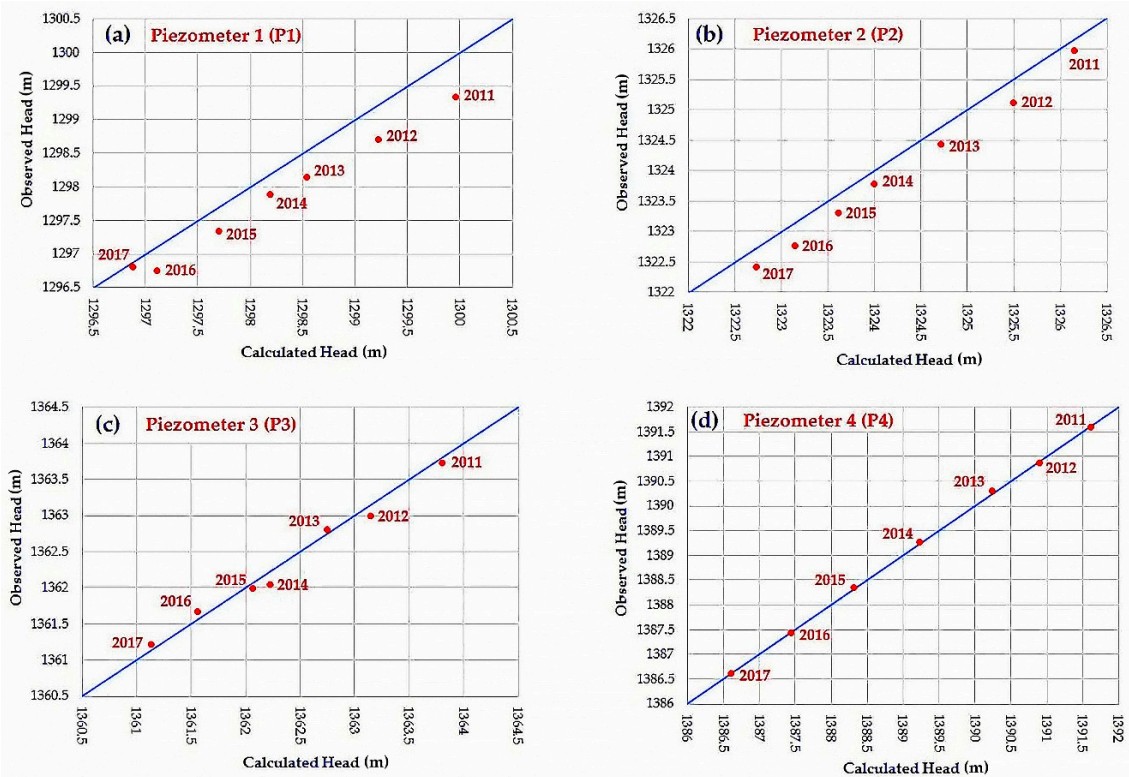

**Figure 17.** Difference between observed and calculated head values in four selected piezometers include (**a**) piezometer 1; (**b**) piezometer 2; (**c**) piezometer 3; and, (**d**) piezometer 4, at the end of model verification for a 7-year period.

## 4. Conclusions

Over-exploitation of the groundwater resources in most plains in Iran is common, and continuing the present pressure on these quite precious sources will lead to the occurrence of severe irrecoverable water stress in the country. In this study, the Birjand aquifer in South Khorasan province, Iran is investigated where the groundwater is the main source of water supply. The study area has been explained extensively. One of the major goals of this study is to improve the accuracy of the aquifer model and to overcome the data shortage, especially in the input boundaries. To do this, a semi-transient approach has been used in order to investigate the aquifer model during a period of time. Coupling MODFLOW with ArcGIS using GMS powerful software allows us to simulate the groundwater flow in the desired area. In the previous studies conducted for Birjand region, we noticed a great difference and disagreement in the considered boundary conditions as a gap of the literature. In the present

research, the comprehensive study in the Birjand region caused some important improvements in the Birjand aquifer model, especially in boundary conditions.

The model calibration is done using steady-state and semi-transient approaches. The aquifer model was investigated for 29 seasons and the result presented. In addition, four piezometers were selected randomly from different parts of the aquifer to comprehensively showing the groundwater level changes in the entire area. To quantify the reliability of the model, some evaluation indices—including mean error, mean absolute error, and root mean square error—are calculated. According to these indices, the performance of the model is promising. The approach was used in this study (i.e., semi-transient calibration) can be applied for other regions with a similar problem as well as similar condition. The findings of this study can improve the status of groundwater resource management in the Birjand region and contribute to the sustainable development of this vital resource.

**Author Contributions:** Conceptualization, A.A.; Formal analysis, R.A.; Methodology, R.A. and A.A.; Project administration, A.A.; Software, R.A.; Supervision, A.A.; Validation, R.A.; Visualization, R.A.; Writing—original draft preparation, R.A.

**Funding:** This research received no external funding.

**Acknowledgments:** The authors thank the Regional Water Company of South Khorasan (RWCSK) for the collaborations on providing the data and the information used in this study.

**Conflicts of Interest:** The authors declare no conflict of interest.

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
