# Peer review of "Application of MODFLOW with Boundary Conditions Analyses Based on Limited Available Observations: A Case Study of Birjand Plain in East Iran"

_water, doi:10.3390/w11091904_

Round 1
Reviewer 1 Report
This manuscript aims to model the Birjand aquifer using numerical software to monitor the groundwater status in the region. To get the practical results, the aquifer boundary conditions are improved in the established conceptual method. The results of the proposed model are in good agreement with the observed data and therefore, the model can be used for studying the water level changes in the aquifer. I carefully read this paper and the content is interesting. However, authors must answer following key comments if they want their paper to be accepted for publication by me. The following comments may help to improve the quality of this manuscript.
General comments:
1. Several published papers indicated for groundwater model’s boundary conditions, which is not new and have been investigated widely in the previous studies (Shen et al., 2017; Wu et al., 2016, 2017, 2019; Liu et al., 2018; Wang et al., 2019; Pujades et al., 2012a, b, 2014, 2017). So, it is not clear what makes the present study unique. What is the gap of literature, what is the effective boundary in the numerical model, what is the main contribution of the study, which is different from others should be clearly clarified? It needs to clearly state the contributions of the manuscript in the introduction section.
References:
l Pujades, E., Ander, L., Carrera, J., Vázquez-Suñé, E., Jurado, A. (2012a). Barrier effect of underground structures on aquifers. Engineering Geology. 145-146(6), 41-49.
l Pujades, E., Carrera, J., Vázquez-Suñé, E., Jurado, A., Vilarrasa, V., Mascuñano-Salvador, E. (2012b). Hydraulic characterization of diaphragm walls for cut and cover tunnelling. Engineering Geology. 125(27), 1–10.
l Pujades, E., Vázquez-Suñé, E., Carrera, J., Jurado, A. (2014). Dewatering of a deep excavation undertaken in a layered soil. Engineering Geology. 178, 15-27.
l Pujades, E., De Simone, S., Carrera, J., Vázquez-Suñé, E., Jurado, A. (2017). Settlements around pumping wells: Analysis of influential factors and a simple calculation procedure. Journal of Hydrology. 548, 225-236.
l Wang, X.W., Yang, T.L., Xu, Y.S., Shen, S.L. (2019). Evaluation of optimized depth of waterproof curtain to mitigate negative impacts during dewatering, Journal of Hydrology, 577(2019), 123969. https://doi.org/10.1016/j.jhydrol.2019.123969
l Wu, Y.X., Lyu, HM, Han, J., Shen, S.L. (2019). Dewatering-induced building settlement around a deep excavation in soft deposit in Tianjin, China. Journal of Geotechnical and Geoenvironmental Engineering, ASCE, 145(5): 05019003. http://dx.doi.org/10.1061/(ASCE)GT.1943-5606.0002045.
l Liu, X.X., et al (2018). Analytical approach for time-dependent groundwater inflow into shield tunnel face in confined aquifer. International Journal for Numerical and Analytical Methods in Geomechanics, 42, 655-673. DOI: 10.1002/nag.2760
l Shen, S.L., Wu, Y.X., Misra, A. (2017). Calculation of head difference at two sides of a cut-off barrier during excavation dewatering, Computers and Geotechnics, 91, 192-202. doi: 10.1016/j.compgeo.2017.07.014
l Wu, Y.X., et al (2017). Semi-analytical solution to pumping test data with barrier, wellbore storage, and partial penetration effects, Engineering Geology, 226, 44-51. doi: 10.1016/j.enggeo.2017.05.011.
l Wu, Y.X., Shen, S.L., Yuan, D.J. (2016). Characteristics of dewatering induced drawdown curve under blocking effect of retaining wall in aquifer, Journal of Hydrology, 539(2016), 554-566. doi: 10.1016/j.jhydrol.2016.05.065
2. The structural and design of GMS software are not clear such as aquifer geometry and boundaries. All information related to the established FEM should be considered. Otherwise, how the authors dealt with the conflicting data that appeared at all the boundary of this aquifer in various published studies, how they control the quality of data obtained from RWCSK, and how we can trust with these established input data?
3. Omitted information related to the geological condition, aquifer parameters, and permeability that used to establish this work. Otherwise, what geotechnical classification have the materials that make up the strata, and based on which they were obtained.
4. Quality of (Figs. 4, 15, and 16) are not qualified for the publication, please revise the journal requirements.
5. I totally disagree with the sentence of statistical error is used to calibrate the established model such as RMSE, MAE, and ME. In my opinion these statistical errors can be used for evaluate the model performance not for calibration. For calibration, the authors need to compare the results with well-known software, or theoretical, empirical or etc.
6. The major goals of this study is to improve the accuracy of the aquifer models, but it is not clear how the authors improved the model, obviously, authors have just applied this software in this region, Pls. clarify.
Abstract
· “in the region” what the authors mean by this region?
· The meaning of this sentence is not clear, “To get the practical results, the aquifer boundary conditions are improved in the established conceptual method”.
· The methodology in this abstract needs to be more clear.
Introduction
· Line 27-28: authors should display general information related to the effective input data that can achieve accurate model.
· Line 10-20: the authors should focus on the main contribution of this work.
· Authors should mention the different models which are qualified for estimating the groundwater status.
· What is the gap of literature in this work?
2. Materials and Methods
2.1. Study Area
· All information in this section are interesting, however, authors should add the geological characteristics of the Birjand Plain area.
· To improve the quality of this section, soil properties and aquifer data need to be added.
· Scale in Fig. 5 need to be added.
2.2. Groundwater modeling of Birjand Aquifer
· It is not clear how the using of very few field operations can achieve very promising results through a mathematical model.
· Line 106-108: what is the difference between these models: Visual MODFLOW, Finite element subsurface flow system (FEFLOW), groundwater modeling system (GMS), Processing MODFLOW for Windows (PMWIN).
· Line 109: what is the difference between GMS software and the previous techniques?
2.3 Groundwater Conceptual Model of Birjand Aquifer
· In the study region there are many challenges such as Lack of adequate knowledge, Lack of adequate and accurate statistics, etc. so how authors avoid all of these challenges.
· Line 156-157: Step by step procedure for developing the conceptual model of the Birjand aquifer should be considered.
· Line 160-165: How the authors established the aquifer geometry of the model?
· Line 173-174: It is not clear how the authors control the quality of data obtained from RWCSK.
· Line 189-191: authors stated that in previous studies carried out there is difference in determining the governing boundary conditions in Birjand aquifer. What is the difference between the previous studies that used geological surveys and this study and how the authors dealt with these differences and avoid it?
· Line 220-239: what is the main properties of soil and Lithology of the established layers.
· The size of Figs. 10 -10 is very big.
· Line 263-264: “It necessarily is mentioned that the distance between the input boundary and adjacent observation well should not be too high”. The technical writing of this sentence should be improved. Otherwise, what is the applicable distance between the input boundary and adjacent observation?
· Line 279-284: all parameters related to the established model should be added. Otherwise, the step procedures of this technique should be added.
· What is the benefit from Fig. 13?
3. Results
· What is the benefit from Fig. 14?
· Why there is a big difference between the observed and predicted data in Figs. 15a, b?
· Section 3.2 should be modified to “Model evaluation”
· Figs. 15-16 are not clear
· Title of Fig. 16 is very long.
4. Conclusions
Conclusions should be clearer. Authors should consider the above mentioned comments.
Reviewer 2 Report
The study entitled “Improving Groundwater Model Boundary Conditions Using Limited Available Conditions, Case Study: Birjand Plain, East Iran” was submitted for publication. The atudy has highlighted to improve the boundary conditions of GW model using MODFLOW. The study can be more scientifically sound if the following is done:
The title must be revised, which will highlight the important findings of thE study. Thus, remove the phase “using limited available conditions, case study”. You can stick with: Improving Grounwater Model Boundary Conditions of Birjand Plain, East Iran. The introduction should be able to discuss about other GW model boundary conditions and discuas the difference of this study to those other models. The introduction should also express justification why you need to improve such boundary conditions. Figure 15 P1, observed head has fluctuating pattern as compared to the linear pattern of the calmulated head using MODFLOW... graphically it was never a good agreement. Express a better explanation on this..consider ground recharge. the authors should also express justification why RMS and not RMSD ot R-squared is used to justifiy agreement between observed and calculates head. RMS values of 0.2 and above in this case, due to limited available data is not a convincing value.Author Response
Please see the attachment.

Round 2
Reviewer 1 Report
The gap from literature on use of Modflow still did not analysed in details.
Author Contributions: Conceptualization, A.A. ; Formal analysis, R.A.; Methodology, R.A. and A.A.; Project
387 administration, A.A.; Resources, A.A.; Software, R.A.; Supervision, A.A.; Validation, R.A.; Visualization, R.A.;
388 Writing – original draft preparation, R.A.
Please combine one auhtor's contribution together.
Reviewer 2 Report
Dear authors,
Congratulations for the substantial revisions. However, regarding the use of RMS,it is not enough justification that it was used by previous studies. You should be able to justify what is the importance of RMS is your data. This is because RMS is usually used for electrical and ECE stuffs. Though it can be used to assess flow of fluids, strong justification is required for using it on ground water systems.
Add a paragraph or 2 in the methods explaining the use of RMS in this system.
